# Is There an Academic Bias against Low-Energy Sweeteners?

**DOI:** 10.3390/nu14071428

**Published:** 2022-03-29

**Authors:** David J. Mela

**Affiliations:** Independent Researcher, 5554 EB Valkenswaard, The Netherlands; djmela@djmela.eu

**Keywords:** sweeteners, research, public health, evidence

## Abstract

This perspective considers evidence of a common academic bias against low-energy sweeteners (LES). The core proposition is that this bias is manifested in research and reporting focused on generating and placing a negative spin on LES, largely through selective citation, interpretation and reporting. The evidence centres on three inter-related points, which together may generate a misleading impression of the balance of evidence: (1) basic and mechanistic research on LES perpetuates “explanations” for unsubstantiated adverse effects of LES; (2) the literature on LES—particularly narrative reviews and commentaries—continually reprises hypotheses of adverse effects without acknowledging where these hypotheses have been rigorously tested and rejected; and (3) negative interpretations of the effects of LES largely rely upon selectively emphasising lower-quality research whilst ignoring or dismissing higher-quality evidence. The expert community should consider these issues in assuring scientific integrity and balance in the academic discourse on LES, and how this is translated into messages for public health and consumers.

## 1. Introduction

There are a large number of nature-derived and artificial sweeteners that can deliver desired levels of sweetness with negligible metabolisable energy. These low-energy sweeteners (LES) include a diversity of high-potency sweeteners (e.g., aspartame, sucralose, saccharin, cyclamate, acesulfame-K, stevia glycosides, mogrosides, thaumatin, or combinations of these), typically used in milligram quantities, and also some bulk sweeteners, such as psicose (allulose) and erythritol.

While the various individual LES differ in chemistry, bioavailability and metabolism, they are all used in technical applications to deliver the intended consumer benefit of reducing intakes of free sugars, with low cariogenicity and glycaemic impact. There are, however, widely differing views about the potential beneficial or adverse impacts of LES on diet and health, based on evidence from in vitro and animal research, observational (cohort) studies, and human intervention trials. In a previously published “perspective” paper, we [1] expressed a number of concerns about the scientific literature on LES, and proposed improved standards for research and disciplined writing on the impact of LES on nutrition and health. In that paper, we were critical of how research on LES was being designed, interpretated and cited; however, we did not explicitly focus on the possible underlying causes for these issues, or how they might bias the field. Nevertheless, it was apparent that the practices of concern were being applied disproportionately to underpin or generate conclusions antipathetic to LES (i.e., promoting putative risks or discounting potential benefits).

## 2. Perspective

The present analysis is not a review, but a perspective that specifically considers patterns of systematic misrepresentation and bias against LES in the scientific literature, manifested in research and reviews placing a negative “spin” on LES through selective design, interpretation and reporting. Three main issues are described and illustrated by examples of evidence from the research literature, supplemented by speculation about the possible reasons for or drivers of such a bias within the academic nutrition and public health communities. The implication is that the scientific community (and popular opinion as collateral damage) is badly served by practices reflecting a widespread bias towards an anti-LES framing of research.


*(1) Mechanisms without relevance.*


There is a long list of mechanisms that have been proposed to explain supposed adverse effects of LES on appetite control, body weight and metabolic health [2,3,4]. These derive support from a large body of research claiming evidence of such mechanisms (e.g., from gut microbiota, neural “taste” responses, and animal feeding paradigms), and imply that these substantiate or explain the adverse effects of LES on glycaemic control, appetite and energy balance in humans. The scientific literature is replete with narrative reviews claiming to address mechanisms “…responsible for the development of metabolic syndrome associated with NNS [non-nutrivie sweetener] consumption” [5] and that “…contribute to the negative metabolic effects of non-nutritive sweeteners…” [6], “…demonstrating the role of the microbiota in glucose intolerance in response to noncaloric artificial sweeteners…” [7]. Indeed, it seems that researchers assume that (new) mechanistic evidence of adverse effects will be found even before experiments are run, pre-registering research protocols with titles, such as “Low-calorie Sweeteners Induce Metabolic Dysregulation Via Alterations in Adipose Signaling” (ClinicalTrials.gov Identifier NCT03125356; an uncontrolled trial that, in fact, found no metabolic dysregulation or changes in any higher-level physiological markers or outcomes [8]).

A key question to ask is: what empirical evidence is there for the outcomes (adverse effects of LES) that these mechanisms propose to explain? Given this extensive literature on mechanisms, one might expect that there is an equally large and convincing body of evidence demonstrating the adverse effects of LES on humans, prompting the need for mechanistic understanding and explanations. Yet, *every* recent systematic review (SR) and meta-analysis of randomised controlled trials (RCTs) in humans—from acute studies of appetite and glycaemic responses through longer-term studies with anthropometric and glycaemic control endpoints—finds neutral or beneficial effects of LES, and almost no indications of adverse effects on these outcomes [2,9,10,11,12,13,14,15,16,17,18,19,20]. Thus, the stream of papers offering a multitude of mechanisms for the adverse effects of LES exists within the context of a wall of empirical evidence that fails to find any adverse effects to be explained.

The implication is that much of the literature postulating the adverse effects of LES has little apparent explanatory value; the research is generating hypothetical pathways to non-existent phenomena. This is not saying that the mechanistic research is faulty or fraudulent, but that it does not have any counterpart in the direct empirical evidence. Worse, much of the mechanistic literature fails to acknowledge this discrepancy, and persuades readers to take evidence of putative mechanisms ipso facto as evidence for the adverse effects of LES. The presumed logic is that, if a mechanism is claimed to lead to adverse effects of LES, LES must have adverse effects—even if this is directly disconfirmed by the empirical evidence. It serves the purposes of authors and the wider anti-LES narrative to perpetuate this misrepresentation, which turns the usual hierarchy of evidence on its head.

An unfortunate corollary of this is that the most convincing and widely cited pieces of mechanistic research on LES may be, paradoxically, most likely to misdirect and mislead. An example of this is the significant impact of research by Suez et al. [21], which proposed that LES-induced changes to the intestinal microbiota lead to adverse effects on glucose tolerance. Despite a number of significant shortcomings in that research [22,23], this single paper set off a stampede of research funding and activity on LES and the microbiota, and left much of the scientific community presuming that the original premise was simply true. Yet, the supposedly explained phenomenon (i.e., LES producing impaired glycaemic control) has no empirical support from comprehensive systematic reviews of the evidence [9,13,14,17,18,24,25]. This fact by itself should render the supposed “mechanism” specious, while years of subsequent research have also left doubt on the replicability of the initial research and basic premise that LES cause important, pathological changes in the gut microbiota [12,19,23,26].

Much of the mechanistic research on LES may simply be documenting the fact that LES-containing stimuli interact with biological systems (which must be true—otherwise they could not be perceived as sweet and pleasant), and with sufficient exposure and effort one may find traces of this somewhere in measures of neural activity, metabolome or microbiota. The bias and spin arises when authors extrapolate from this to create a more compelling and provocative story, forgetting that sensing and signalling do not equate to salience and significance.

It is mainly the associations observed in some observational studies that, with their repeatedly articulated limitations [27,28,29,30], lend credence to the search for mechanistic explanations for supposed adverse effects of LES, since the support is not clearly coming from the controlled trials. The result is an absurd situation in which mechanistic and observational research continually reinforce each other in generating hypotheses that consistently fail to be supported by direct empirical testing.


*(2) Ignoring the rejected hypotheses.*


A “zombie” in science is an idea or hypothesis that has been killed, yet never really dies, and continually rises to live again (adapted from [31]).

For LES, much of the basic research and commentary text is only justified by keeping zombies alive, actively ignoring large swathes of research as if it never existed. How else is it possible to still repeat the view that exposure to LES stimulates subsequent increased appetite and energy intake and adversely affects glycaemic control, e.g., [32]? These narratives can only be constructed on a foundation of a narrow selection of basic research and prior opinion, “confirmed” by uncritical reference to observational associations and mechanistic studies, further supplemented by speculation and assumptions. Meta-analyses using data from tens of human trials consistently show that exposure to LES reduces total energy intake vs. caloric sweeteners, and generally has neutral effects on total energy intakes relative to unsweetened (mainly water) controls [10,11,15,16]. Similarly, systematic reviews from a large body of human trials show that LES on their own or together with nutrient loads have no significant effect on subsequent glycaemic or insulinaemic responses [9,14,17]. This is not to say that LES are necessarily beneficial, but that the empirical evidence, taken as a whole, repeatedly fails to validate the suggested adverse effects. These are trials and outcomes that are not particularly difficult to test and replicate; indeed, they have been, many, many times, making it even more remarkable that these zombies continue to have life breathed into them. 

The persistent repetition of dubious or disproven narratives is not benign. It diverts funding and effort away from more valid research questions, and perpetuates a misrepresentation of the state of science. For example, there is a widespread belief that the exposure to sweetness from LES generates or enhances a (generalised) preference for sweetness, especially in children. This underpins the view that, from at least two World Health Organization regional offices, the replacement of sugars by LES might be counter-productive for public health initiatives [33,34]. However, this popular view is not supported by evidence-based assessments from other public health agencies (e.g., [35]), nor by the only SR of the evidence [36] or further research since then [37]. While future research and reviews could establish a clearer consensus, this looks likea seriously wounded hypothesis, a weak and potentially counter-productive foundation for public health guidance. It is an attractive story that fits nicely into an anti-LES narrative, but stands in the way of evidence-based policy that considers the possibility that LES might satisfy a pre-existing preference for sweetness, rather than drive it. 


*(3) Giving priority to lower-quality evidence.*


In most spheres of evidence-based public health, greatest weight is placed on the highest quality, most direct and generalisable information. Negative scientific spin and anti-LES bias relies heavily on ignoring or downgrading the strongest and most relevant evidence, whilst emphasising a selected subset of less robust and lower-quality sources of information.

Continued concerns about the nutritional implications of LES are largely driven by reference to observational research, and a selected minority of experiments in animal models. In a number of narrative reviews, little distinction is made in the weight given to animal (or even in vitro) studies vs. human intervention trials in drawing conclusions, e.g., [38,39], even for outcomes where repeated human trials fail to support the putative implications of more basic research. We [1] described how selected animal and observational studies could equally be used to underpin an intentionally absurd argument for water as a cause of weight gain and mortality. 

Observational studies (and the reviews citing them) lean toward finding adverse associations of LES with metabolic health outcomes, whereas RCTs lean toward neutral or beneficial effects, often for the same outcomes, such as body weight [40,41]. Large, well-designed observational studies with appropriate analyses can highlight legitimate areas of concern, although there is considerable debate around the use and limitations of observational data in nutrition [30]. For observational research on LES specifically, the validity of certain associations must however be questioned when they are repeatedly disconfirmed by evidence from intervention trials, combined with a susceptibility to confounding that undermines the assignment of a causal interpretation [27,29,42,43]. It is perhaps useful here to consider the “white swan” metaphor [44]: given the chequered history of claims based on nutritional epidemiology, the practice of continuously propping up a particular observation (“all swans are white”) with further observational research is fundamentally undermined when controlled studies reveal something different (i.e., the presence of “black swans”). 

A common feature of anti-sweetener narratives is therefore to ignore or dismiss the more obvious conclusions that would be drawn from the totality of evidence, and fall back on the selective citation of a particular subset of research, particularly animal studies. In truth, however, frequently cited experiments suggesting adverse effects of sweeteners on the body weight and metabolism in rodents, e.g., [21,45], have been unrepresentative of the total body of evidence in animals, and poorly replicable [46,47,48]. SRs of that literature show that a clear predominance of animal studies have found reduced or no effect of either pre- or post-natal exposures to LES on body weight [15,49]. Thus, not only do animal studies have many shortcomings [46], but, similar to human intervention studies, when that evidence is considered as a whole, it also does not support the supposed adverse metabolic effects of LES. 

It is also possible to generate misleading conclusions, even when considering the much better evidence from studies of humans. In a recent citation analysis, we found that the conclusions of reviews on LES and body weight outcomes were not consistently related to the conclusions of the literature cited by those same reviews [41]. Thus, the reviews might conclude “X”, even though the cited evidence said “Y”. Remarkably, in a substantial number of cases, authors of the reviews cited support from papers that did not even address the question. It is easy to dismiss this as an inherent risk of narrative reviews. However, we noted that SRs are also not immune to citation bias, which arises from the application of specific in- and exclusion criteria. This can lead to marked differences in the conclusions of good-quality SRs assessing the same diet, lifestyle or clinical interventions [50,51,52]. 

For LES, the prominent WHO-commissioned SR by Toews et al. [18] presents an extreme example of excluding relevant data. In that SR, authors excluded the largest and most ecologically valid intervention studies—those using real products—by applying an apparently post hoc criterion requiring that the specific LES was identified. This criterion was not specified in the registered protocol, is only really needed for the sub-analyses of specific sweeteners, and of course could never be applied to the (included) analyses of observational studies. Comparing LES to all comparators together (water, caloric sweeteners, nothing, and placebo) further muddies the water, allowing for the pejorative conclusion that LES are “ineffective”, when it is clear that any beneficial effects of LES would mainly derive from reductions in intakes of energy from sugars [20]. An ostensibly high-quality SR can therefore tick all the boxes for its procedures and reporting, yet still introduce bias and misleading results, based on the dubious operational specification of the research question and a poor understanding of the topic matter (see also https://www.bmj.com/content/364/bmj.k4718/rapid-responses, accessed on 29 October 2021). Other advanced analyses of the literature (e.g., umbrella meta-analysis or SR of SRs on LES) could perhaps help assess the extent to which the results are influenced by specific variations in the stated or implied research hypotheses and study selection criteria. 

## 3. The Counter-Arguments

The preceding text is, to some extent, intentionally provocative. Nevertheless, much of what has been described in the present paper reflects poor scientific practice and judgment, underpinned by intentional or unwitting bias, with consequences for scientific understanding, public health guidance and consumer opinion. Some practices, such as giving more weight to mechanistic research than empirical evidence, seem difficult to defend. Nevertheless, there will be a number of counter-arguments to the overall position taken here, with lesser or greater degrees of merit.


*(1) The precautionary principle.*


The precautionary principle in public health emphasises caution where there is a potential for significant harm, and conclusive scientific knowledge is lacking. For food and food ingredients, it is usually considered on the basis of a risk:benefit analysis. It has been argued that this principle should be applied to LES, e.g., on the basis of observed adverse associations with a risk of diabetes [53]. In contrast, we [1] previously argued that the precautionary principle cannot so simply be applied to LES. LES can facilitate the reductions in sugar intake, particularly in beverages, and have been shown to benefit weight control when used in this way. There is direct evidence for the neutral or beneficial effects of LES in human trials, as well as extensive evaluations of their safety. This can be contrasted with the weaker, speculative and uncorroborated evidence of potential adverse effects, extrapolated from mechanistic and observational research. It is therefore debatable whether the precautionary principle as such applies, and may even be argued that this principle would favour promoting rather than discouraging the use of LES as a public health measure. 


*(2) There is not enough long-term research.*


The argument is frequently made that there is insufficient long-term research on LES, though this is never accompanied by any concrete proposal or justification for a duration or design of research that would be sufficient. There have been a substantial number of controlled intervention trials with LES exposures over periods of at least 12–18 months [3]. The results of these longer trials have been largely consistent with each other and, importantly, also consistent with shorter trials, mainly finding neutral or beneficial effects on weight control and metabolic outcomes, in both adults and children [15,16]. While it is plausible that any beneficial effects (as well as compliance) may diminish over time, it is not clear what evidence, hypotheses or other examples in nutrition support the likelihood of a reversal of direction toward an adverse effect. One suspects that if even longer trials fail to show adverse effects, it will be claimed that they *still* were not long enough. However, it seems worth noting that many of these same human trials, using an LES as a comparator, are widely accepted and cited as convincing evidence for the adverse effects of sugars. 


*(3) Industry bias in LES research.*


This claim is usually raised where research (1) has funding or author links to industry, (2) did not produce the assumed “correct” or preferred conclusions, and (3) cannot really be faulted on objective indicators of research quality. The nutrition literature is replete with research testing for potential bias in relation to source of funding. For LES specifically, Mandrioli et al. [54] reported a much greater proportion of favourable conclusions from (mostly narrative) reviews of LES, where the review or authors had even tenuous links to any sort of industrial sponsorship vs. not. As there were no differences in the objective measures of risk of bias, the authors drew a causal interpretation that “…sponsorship and authors’ financial conflicts of interest introduced bias affecting the outcomes”. It is easy to assume that this means that the research conclusions were wrongly manipulated in some way, and from this to the view that any LES research with links to industry support should be downgraded or disregarded. On the other hand, “bias” is used here simply to note a difference in reported results, but says nothing about validity or cause. An objective analysis of primary research trials indicates no difference between industry-funded and non-industry-funded studies in the quantitative effect of LES on body weight outcomes [16]. 

Using the presence or absence of presumed financial interests as a proxy measure for research integrity or validity is by itself a weak and prejudicial research shortcut, an unreliable alternative to careful expert assessment of the research itself. It implies there is some spectre of influence that cannot be detected by objective tools and domains for assessing risk of (e.g., procedural) bias. For reviews, a simple comparison of results or risk of bias measures does not directly inform about the relevance or strength of the hypothesis or evidence base used in addressing the research questions. In the case of LES, one might better consider, for example, whether different funding streams are associated with different in- or exclusion criteria, and emphasis on human RCTs vs. observational, animal and putative mechanistic evidence. A key argument here is that bias largely arises from the differential inclusion and weighting of evidence, and that an anti-LES narrative largely arises from a greater reliance on highly selected and lower-quality forms of evidence. 

A number of potential routes of industrial sponsor influence have been described [55], as have a range of specific misleading practices used in research on “functional” foods and ingredients (Table 2 from [56]). These issues are certainly not exclusive to industrially funded research, which can be “…of equal or better quality than those with other funding sources” [57]. Practices including “p-hacking” and “HARKing” are well-known and widespread in academic research, and provide the tools through which the nature of the results, interpretation and “spin” of otherwise well-designed and executed research can be manipulated by naivety or intent [58,59]. All of these are, however, subject to expert evaluation, and allow objective, transparent means for identifying risk of bias, independent of guilt-by-association judgments on the basis of funding sources. 

In short, “Evidence should be evaluated based on the science, not the scientists” [30]. Arguing that evidence can be dismissed or downgraded simply because of links to industry is a poor substitute for more rigorous, objective analyses.


*(4) Do not need sweeteners.*


LES may be viewed as an unnecessary technological fix to a public health problem brought about by the agro-industry and the proliferation of inexpensive, sugar-sweetened foods and beverages. Importantly, this position is not saying that LES themselves have adverse health effects, but reiterating that any benefits of LES largely reflect the extent to which they help reduce the adverse effects of sugars. This view is critical of LES not because they are inherently bad, but because they should be unnecessary; consumers can be simply advised to drink plain water and eat unsweetened foods. With good compliance, this public health approach would achieve the desired goal of reducing sugar intakes, whilst side-stepping any further scientific consideration of the beneficial or adverse effects of LES. It raises several questions though, mainly whether a focus on avoiding sweetness is the most effective way to achieve sugar reduction goals. Similar questions arise around the use of low-sodium salt replacers. 

It is possible, but debatable, whether pursuing a change in gustatory perception or preferences is realistically possible, necessary or effective as a population strategy. As noted, however, the opposite is also possible, where LES enhance consumer compliance and achievement of dietary goals, by safely satisfying inherent desires for particular tastes [60,61]. 


*(5) Gaps and remaining uncertainties about LES.*


There are outstanding research questions, particularly those that are not readily amenable to direct testing, such as LES exposures during pregnancy and lactation. In contrast, as noted, other research questions have been the subject of considerable direct testing, which yielded consistent results. The implication is that we need *less* research on those, allowing limited resources to be directed to high-quality, hypothesis-led research on the most important gaps. 

## 4. Why Is There a Bias against LES?

Lastly, one may rightly ask why an anti-LES bias might be so widely present in the research community. There are some possible forces that might promote an anti-LES orientation, but this is pure guesswork. The suggestions below are not mutually exclusive, may not be conscious or intentional, and certainly will not apply to all experts or researchers critical of LES. 

White hat bias

“White hat bias” has been defined as “bias leading to distortion of research-based information in the service of what may be perceived as righteous ends” [62]. The corollary of this is belief in a particular underlying truth that supersedes and gives justification for downgrading evidence that might obscure or conflict with this world view. There are few rewards for academics in food and nutrition who are seen to take the “industry” position, and indeed penalties for those who deign to collaborate with commercial food companies (such as a risk of social stigma, charges of bias, and exclusion from membership of expert committees). At the very least, there is a widespread view in the nutrition and public health communities that one must express suspicion if not open antagonism toward research or foods and ingredients supported by commercial interests, and the scientists linked to them. 

Non-financial conflicts of interest:

Not all conflicts of interest involve direct, personal financial reward. There clearly are academics whose research programmes and (non-industry) funding streams, career advancement, publications, speaking invitations, and public and social media presence benefit from propounding a particular, often provocative standpoint. These can act as a highly compelling force to maintain views that have become intertwined with personal reputation, professional position and ego. It is difficult to avoid some degree of “allegiance bias” or “intellectual conflict of interest”, favouring the attachment and defence of one’s own work, preferred theories and points of view [63,64,65]. Furthermore, LES is a topic that continues to attract significant research funding, whether or not one really believes there is an issue to be resolved, or that further research will shift the evidence base on key outcomes. It is cynical but true to say that additional funding comes from proposing provocative new aspects to explore, not from supporting the status quo or suggesting that a research question has been adequately addressed. 

Opposition to (ultra-)processed and reformulated foods

Public health nutrition guidance usually features aspects of both “change food” and “change people” [66]. The use of LES and reformulation are part of a “change food” approach to achieving dietary goals, requiring minimal changes in consumer behaviours. This approach emphasises improving the nutritional profile of a typical basket of commercial foods by the technical reformulation and marketing promotion of healthier alternatives within existing, popular food categories, preferences and habits. In other words, offering the same types of foods, but with a different nutritional profile. In contrast, a “change people” approach focuses on encouraging larger sustained shifts in consumer behaviours, towards other food categories and choices, other preparation and cooking methods, and perhaps also changes in taste preferences.

In recent years, there has been a growing literature on the classifications of foods according to the extent and purpose of food processing, and use of these in dietary guidance. There are number of such classification systems, of which NOVA is the most widely known and used [67]. Within this system, reformulated products using ingredients, such as LES, are criticised and placed within the “ultra-processed foods” category, which has been associated with poor diet quality and health risks [68]. The merits (or not) of these classification systems have been debated [67]; however, in this case, the criticism of LES is not expressing judgment about their safety or efficacy per se, but about their use in making foods more palatable and cheap. This objection to LES and the wider anti-technology bias is underpinned in part by perceived food values, although having foods taste worse and cost more is easy to accomplish and might indeed reduce overconsumption.

## 5. Conclusions

In conclusion, I would reiterate that this is a particular perspective on the field. It is not a systematic review, and my own biases are self-evident and acknowledged. I am, however, less concerned with where the evidence on LES takes us, and more concerned with the nature of the research process, and how the resulting evidence is being interpreted, assessed, reported and applied. The argument here is *not* that LES are beneficial for health, but that there is a systematic bias toward presenting them as bad for health, underpinned by the selective and misleading use of evidence. There will be others who disagree with the core proposition and examples, and constructive responses to this perspective should be welcomed. The points made in an earlier commentary [1] address the issues relevant to the entire field, regardless of one’s current views about LES, and application of the practices recommended there could address many of the issues raised in the present paper. We ultimately need a community of experts with an extensive knowledge of the subject, who carefully evaluate the evidence and bring an dispassionate balance to the scientific and public discourse on LES. Some may reach the conclusion that the risks of LES outweigh the benefits, or that the potential risks or gaps in evidence raise a significant cause for concern. Others may come to a different or opposing view. Whether we like or dislike LES, the focus should be on achieving public health goals, based on the best quality, totality and weighting of evidence.

## Data Availability

Not applicable.

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
