# Peer review of "Is There an Academic Bias against Low-Energy Sweeteners?"

_nutrients, 2022, doi:10.3390/nu14071428_

Round 1

Reviewer 1 Report

This perspective review provides a provocative analysis of what appears to be an "academic bias" against low-energy sweeteners (LES) by a food industry scientific expert on the topic. The role of LES in human food intake and health has been extensively studied and reviewed in academia and is the subject of much interest in the public health field and popular press. One reason for the renewed interest in this topic during the last 15 years were the important discoveries that sweet taste receptors exist outside the mouth in organs related to digestion, metabolism, satiation, and reward (stomach, intestine, pancreas, brain). This led to many investigations as to how LES might act at postoral sites to influence food intake, digestion, metabolism, etc. I suspect that part of the bias to report potential negative health effects of LES at these sites is because this is more "newsworthy" and worthy of grant support than reports of no negative effects. Dr. Mela has provided a thoughtful and well-researched review of this topic which should stimulate much interest and discussion.

I have only one minor recommendation.

LES should be described as "high-potency" rather than "intense" sweeteners in the Introduction. The term "intense sweeteners" is misleading because LES are not much sweeter than natural sugars (Antenucci & Hayes, Int.J.Obes. 2014, 39, 254-259).

Author Response

“Dr. Mela has provided a thoughtful and well-researched review of this topic which should stimulate much interest and discussion.”

Response: I hope this will stimulate discussion and perhaps resolution of some of the issues, and have added text specifically inviting constructive commentary from those with other views.

“LES should be described as "high-potency" rather than "intense" sweeteners in the Introduction. The term "intense sweeteners" is misleading because LES are not much sweeter than natural sugars (Antenucci & Hayes, Int.J.Obes. 2014, 39, 254-259).”

Response: This is a valid point of concept and terminology, and has been adopted in revision.

Reviewer 2 Report

This is a paper highlighting the potential “academic bias against low-energy sweeteners”.  While it is well-written, I have some comments/ concerns:

  • This is a very subjective opinion for a narrative review. Why not conduct an umbrella meta-analysis or a systematic review of the systematic reviews to be more objective?
  • Why not use a study quality measure and rate the quality of these systematic reviews or meta-analyses (GRADE criteria)?
  • This paper is very one-sided and downgrading observational studies even though there are strong and valid conclusions that were drawn from observational data (claims; electronic health records). There are longitudinal cohort studies with excellent quality and large sample sizes and a great information on many confounders.  Such analyses can shed greater light on this legitimate concern about artificial sweeteners.
  • Rather than taking side towards the benefits of LES, I suggest a more objective paper and approach.
  • Conflict of interest. The inherent bias while writing this paper is tangible and predictable.

Author Response

“This is a very subjective opinion for a narrative review.”

Response: The manuscript was not presented as or intended to be a narrative review of the topic. It appears that the text and guidance from the journal were not sufficiently clear in highlighting that this was submitted as a “Perspective”, and as such represents the author’s personal assessment of the field. Nevertheless, the reviewer has not highlighted any specific errors in the stated facts or examples. For clarity, the revised text better emphasises the nature of the article type, specifically that it is a personal perspective/proposition, not a review, and constructive commentary is welcomed from those with other views.

"Why not conduct an umbrella meta-analysis or a systematic review of the systematic reviews to be more objective?

Why not use a study quality measure and rate the quality of these systematic reviews or meta-analyses (GRADE criteria)?”

Response: Text has been added to note that these types of analyses might provide further insight on the topic; however, actually conducting these analyses is clearly outside the scope of this article type. These would be substantial original research activities, requiring a significant effort from multiple expert co-authors. Furthermore, the manuscript makes the point that systematic reviews and meta-analyses are also subject to biases, e.g. related to choice of hypotheses, particular in-/exclusion criteria and comparators – which require expert judgement and are not really captured by existing, standard quality assessment tools.    

“This paper is very one-sided and downgrading observational studies even though there are strong and valid conclusions that were drawn from observational data (claims; electronic health records). There are longitudinal cohort studies with excellent quality and large sample sizes and a great information on many confounders.  Such analyses can shed greater light on this legitimate concern about artificial sweeteners.”

Response: The text has been revised to acknowledge that observational studies may highlight legitimate areas of concern, but within the context of a wider debate on the value (and limitations) of observational studies in nutrition. For observational studies of LES, it is an unavoidable fact that the validity (causal interpretation) of certain associations is badly undermined by repeated disconfirmation in intervention trials, combined with the existence of good alternative explanations for observed associations (e.g. reverse causality).

“Rather than taking side towards the benefits of LES, I suggest a more objective paper and approach.”

Response: The core proposition is not that LES are beneficial per se, but that there is a widespread systematic bias toward presenting them as bad for health. This point is reiterated in the conclusions. The text makes it very clear that there are differing points of view on LES, with extensive reference to key research hypotheses, mechanisms and papers that present LES in a negative light. In underpinning my perspective, it is appropriate to convey the counter-arguments to these, including evidence indicating neutral or beneficial impacts of LES. The revised text further emphasises that this is a personal perspective, not a review.

“Conflict of interest. The inherent bias while writing this paper is tangible and predictable.”

Response: This point is explicitly acknowledged in the manuscript (“my own biases are self-evident and acknowledged”), and is also inherent in the “perspective” article type, a personal assessment of the field. This should be clear to the audience, and it’s not clear what further action is recommended. The revised text further reiterates that this a personal perspective, and invites constructive commentary from those with other views. A point made in the manuscript is that conflicts of interest are almost unavoidable: Supposedly “independent” academics have a wide range of non-financial and intellectual conflicts that may bias research interpretation and reporting, but which are rarely acknowledged.